# Icing Time Prediction Model of Pavement Based on an Improved SVR Model with Response Surface Approach

Lingxiao Shangguan [1], Yunfei Yin [1,*], Qingtao Zhang [2], Qun Liu [2], Wei Xie [2] and Zejiao Dong [1]

1   School of Transportation Science and Engineering, Harbin Institute of Technology, Harbin 150090, China
2   Shan Dong High-Speed Construction Management Group Co., Ltd., Jinan 250101, China
*   Correspondence: yyfeillei@gmail.com or yunfeiyin@hit.edu.cn

**Abstract:** Pavement icing imposes a great threat to driving safety and impacts the efficiency of the road transportation system in cold regions. This has attracted research predicting pavement icing time to solve the problems brought about by icing. Different models have been proposed in the past decades to predict pavement icing, within which support vector regression (SVR) is a widely used algorithm for calibrating highly nonlinear relationships. This paper presents a hybrid improved SVR algorithm to predict the time of pavement icing with an enhancement operation by response surface method (RSM) and particle swarm optimization (PSO). RSM is used to increase the number of input data collected onsite. Based on that, the optimal SVR model is established by optimizing the kernel function parameters and penalty coefficient with the particle swarm optimization (PSO) algorithm. The hybrid improved SVR is compared with SVR, PSO-SVR, and RSM-PSO for coefficient of determination ($R^2$), mean absolute error, mean absolute percentage error, and root mean square error to check the effectiveness of PSO and RSM in optimizing SVR. The results show that the combination of two methods in the hybrid improved algorithm has a better optimization capability with $R^2$ of 0.9655 and 0.9318 in a train set and test set, respectively, which outperforms PSO-SVR, RSM-SVR, and SVR. In addition, the $R^2$ of the hybrid improved SVR and PSO-SVR both reach the optimal fitness value approximately at the iteration of 20, which suggests that convergence capacity remains relatively constant with the predictive accuracy being improved.

**Keywords:** icing prediction; support vector regression; particle swarm optimization; response surface method

## 1. Introduction

Pavement icing constitutes a main factor impacting the driving safety and the efficiency of road transportation systems because the formation of ice on pavement texture induces the dramatic decline of friction coefficient of pavements, especially for cold regions with long winters. How to solve this problem has attracted research in different fields, and many methodologies or practice have been proposed in the past decades. Over the years, one of the commonly used practices applied to alleviate this adverse effect is to apply salt after the ice is formed on the pavement due to the convenience of application, though it needs four to ten times the salt to remove the ice from the pavement than to prevent its formation by pre-salting. Moreover, previous studies have found that the excess use of salt induces the deterioration of the roadside environment, such as soil pollution, death of some buds, and delay in growth of trees [1–3]. Additionally, salting is also an important contributor to the corrosion of pavement structures, which is unfavorable for the maintenance of roads [4,5]. Recently, one alternative, icing sensor, has appeared as a new approach to monitor the formation of ice and guide the treatments for the pavement icing problem. This approach distinguishes the freezing conditions based on the different properties of ice, water, and air to monitor the formation of ice. It can be divided into contacting and non-contacting type with a wide application in pavement structures [6]. Due to its real-time feature, it is

considered as an effective technique in pavement icing prediction, while the accuracy and cost of the sensor, as well as the difference between road construction material and sensor equipment, still demand further research. Thus, it appears necessary to propose different approaches to solve this problem.

Recently, different icing prediction models have been proposed. In order to build an accurate prediction model, researchers investigate the impact of different environmental factors on pavement icing. It is suggested that pavement temperature, temperature, dew point, relative humidity, wind speed, and precipitation are commonly considered as the main factors in the process of freezing [7,8]. Based on that, highway engineers and meteorologists have adopted numerical models to predict the formation of ice with advance knowledge of where and when ice will occur. This technique can provide valuable information for winter maintenance services with a good performance in different countries. However, most numerical models are empirical due to the complex properties of ice and this may not reflect the real ice formation process without an adequate database [9]. The disadvantages have hindered its usage in regions without complete systems to collect climatic information over the long term. The support vector machine (SVM) is a promising technique for small sample prediction. In the past research, it has been combined with an optimization method of differential evolution to forecast the road icing, and shows feasibility and effectiveness [10]. Besides that, different environmental factors that affect the freezing of pavements are considered to build the road icing prediction model by support vector classification (SVC), and the model can predict the pavement icing warning accurately [11]. While most published studies focus on the judgment of pavement icing conditions, the prediction of pavement icing time is seldom involved. Derived from SVM, support vector regression offers unique advantages over nonlinear regression analysis. It has been verified that the SVR model is valid in different areas, such as air pollutant concentration prediction [12], soil internal friction angle [13], and prediction of remaining service life of pavement [14]. The results in past research have demonstrated that SVR can predict the nonlinear problems with high accuracy and robustness [15]. This algorithm calibrates the relation of input and output variables by transforming the nonlinear problem in a low-dimensional space into high dimension and finding a hyperplane to fit the relation between the input and output. Commonly used kernel functions of SVR include linear kernel functions, polynomial kernel functions, radial basis kernel functions, and sigmoid kernel functions [16]. Gaussian radial basis kernel function is the most widely used kernel in SVR and it has been applied in related fields [17]. In SVR, insensitive loss coefficient $\epsilon$, penalty parameter $C$, and parameter of kernel function directly affect the accuracy of prediction, while obtaining the optimal parameter combination is usually time-consuming. Therefore, it demands a proper optimization algorithm to enhance the accuracy of prediction.

Numerous optimization algorithms with different applications have been developed by researchers in the published studies [18–20]. The grid search (GS) algorithm tests all possible parameter values to optimize and compares the result of each parameter combination for the optimal solution [21]. Numerous attempts guarantee the effectiveness in optimizing the penalty parameter and kernel function parameter, leading to high time consumption as well. Thus, research has been conducted to reduce the time cost by automatically changing the search range and step [22]. The genetic algorithm (GA) method is another optimization algorithm imitating the revolution of creatures and is applicably administered to determine optimal SVR parameters [23]. However, the genetic variation and crossover in GA cannot be controlled, which may induce poor prediction capability at times. Swarm intelligence [24], a subset of artificial intelligence (AI), has gained more attention, as more high-complexity problems require the acquisition of optimal solutions that are not achievable within a reasonable time by previous methods. It can be divided into insect-based algorithms and animal-based algorithms, according to the previous research. Among these algorithms, the firefly algorithm (FFA) has exhibited promising accuracy in monthly rainfall forecasting [25]. The main parameters of FFA, including $\alpha$, $\beta_0$, and $\gamma$, unavoidably affect the result of optimization, and the determination of optimal parameter

combination is costly; thus, applying FFA in an SVR model may not be quite favorable. Besides FFA, the particle swarm optimization (PSO) algorithm is another popular and straightforward optimization methodology, inspired by the collective behavior of social animals [26]. The inertia factor of PSO is the major factor affecting the result of optimization, thus this algorithm has been applied in nonlinear problem prediction and demonstrates good performance [27]. To avoid the local optimal solution, different methods are proposed to change the inertia coefficient in the process of iteration [28]. Besides SVR parameters, the input data is another factor dominating the accuracy of prediction. Thus, it is necessary to calibrate the input data for modeling icing time prediction. To obtain better accuracy, response surface methodology (RSM) is used in this paper for the optimization of input variables. RSM is a collection of mathematical techniques based on the fit of equation to experimental or real statistics [29], which describes the relation between response and input data. In recent studies, RSM has been gradually utilized to optimize the input variables for modeling SVR [30]. The prediction accuracy of SVR is greatly promoted after RSM is utilized to calibrate the input variables, which shows superiority to other methodologies [31]. The commonly used SVR prediction models are summarized in Table 1.

**Table 1.** Different SVR prediction models.

| SVR Model | Feature | Limitation |
|:---:|:---:|:---:|
| SVR | The model parameters are set manually | To obtain the optimal is costly |
| GA-SVR | Optimize the parameter combination by simulating the genetic variation and crossover | The optimization of prediction capability is not always favorable |
| GS-SVR | Optimize the parameter combination by searching every possible solution | The optimization is time-consuming |
| PSO-SVR | Optimize the parameter combination by imitating the collective behaviors | The local optimal solution may affect the prediction capability |
| FFA-SVR | Optimize the parameter combination by imitating the accumulation of fireflies | The determination of parameters for FFA increases the computation consumption of prediction model |
| RSM-SVR | The prediction accuracy is improved by expanding the number of input variables | Same limitation as SVR |

This paper develops a hybrid improved SVR model for the prediction of pavement icing time by combining the advantages of RSM and PSO when calibrating SVR. The collected environmental data, including initial pavement temperature, average pavement temperature, average circumstance temperature, humidity, radiation, wind speed, and water film thickness, are initially calibrated by RSM to provide the input variables for SVR. After the first calibration process, the input variables in the train set are combined with corresponding icing time to model SVR using Gaussian radial kernel function, predefined penalty coefficient $C$, and kernel function parameter $\gamma$. Considering the simplicity of the PSO algorithm, it is used to obtain the optimal parameter combination of penalty coefficient $C$ and kernel function parameter $\gamma$ for the SVR model established in the last step. Finally, the experimental results of the hybrid improved SVR are compared with SVR, PSO-SVR, and RSM-SVR in $R^2$, $MAE$, $MSE$, and $MAPE$ to verify the performance of RSM and PSO. The results demonstrate that the hybrid improved SVR outperforms SVR, PSO-SVR, and RSM-SVR in $R^2$ of train set and test set, respectively, 0.9655 and 0.9318, indicating that the combination of PSO and RSM shows better performance in enhancing the prediction capability of SVR. In addition, the hybrid improved SVR and PSO-SVR both converge approximately at the iteration age of 20, suggesting that the prediction accuracy of the proposed model increases as the convergence speed stays relatively constant.

The structure of this paper is arranged as follows. Section 2 illustrates three components of the hybrid improved SVR, including SVR, RSM, and PSO, with the framework of

the proposed hybrid model diagrammed by figure. Based on that, four prediction models, SVR, PSO-SVR, RSM-SVR, and hybrid improved SVR, are calibrated in Section 3, and the accuracy and convergence speed are used to verify the performance of PSO and RSM. Finally, the conclusions are drawn in Section 4.

## 2. Modeling Method

In this paper, the commonly used SVR model is optimized by an improved PSO and polynomial RSM to calibrate the high-accuracy icing time prediction model, using the data collected onsite. The collected environmental data are initially calibrated by RSM and split into a train set and test set by cross validation (CV) method for the construction of SVR with predefined parameters including $\varepsilon$, $\sigma$, and $C$. The parameter combination of $\sigma$ and $C$ is updated in the iteration of the improved PSO, and the optimal solution is found to build the hybrid improved SVR model. The whole procedure is diagrammed in Figure 1.

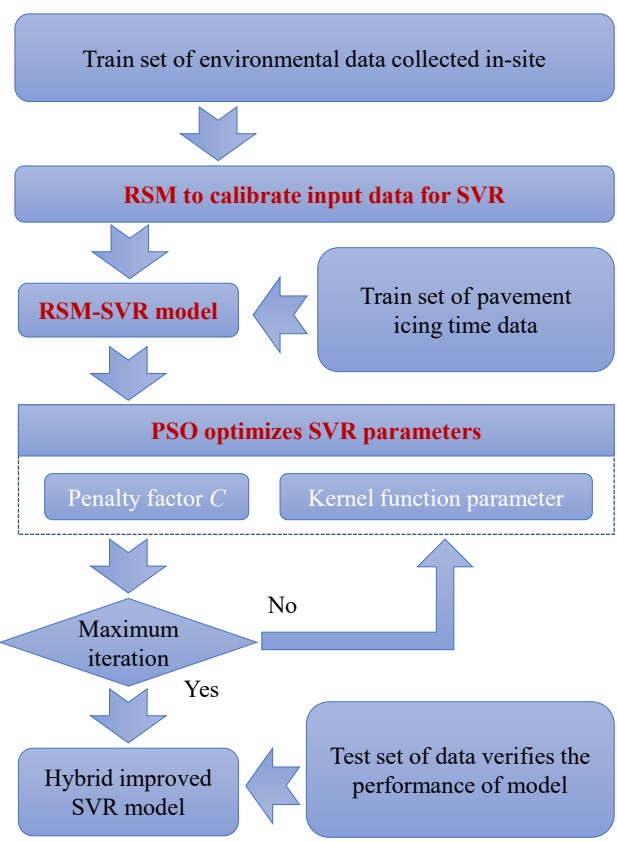

**Figure 1.** The framework of the hybrid improved SVR.

### 2.1. Support Vector Regression

SVR is derived from support vector machine (SVM) and is usually used as classification methodology. This model converts the nonlinear problem in low-dimensional space into high-dimensional space and finds a hyperplane to fit the relation between input and output. Researchers have developed different kinds of SVR models in the previous studies. Among these types, $\epsilon$-SVR is considered as the most applied type and is utilized in this paper with an introduction in the following paragraphs. Given a training dataset $D = \{(\mathbf{x}_i, y_i), i = 1, 2, \ldots, m\}$ with $m$ samples, where $\mathbf{x}_i$ and $y_i$ represent input vector and actual value respectively, a regression function can be built to approximate the target value $y_i$,

$$f(\boldsymbol{\chi}) = \boldsymbol{\omega}^T \phi(\boldsymbol{\chi}) + b, \tag{1}$$

where $\chi = (\mathbf{x}_1, \mathbf{x}_2, \dots, \mathbf{x}_m)^T$ refers to $m$ groups of input vectors, $\phi(\chi)$ transforms the sample from Euclidean space to high-dimension space, $\omega$ is a weight vector, superscript $T$ denotes transpose, and $b$ is a bias value. $\varepsilon$-SVR method adopts a new type of loss function, which creates an insensitive zone by maximum $\varepsilon$ value. $\varepsilon$ refers to the maximum deviation from optimal hyperplane. When the absolute value of deviation between hyperplane function $f(\chi)$ and target $\mathbf{y} = (y_1, y_2, \dots, y_m)^T$ is less than $\varepsilon$, the loss is acceptable. Based on that, the SVR problem can be formalized as

$$\min_{\omega, b} \frac{1}{2}\|\omega\|^2 + C\sum_{i=1}^{m} l_{\varepsilon}(f(\mathbf{x}_i) - y_i), \tag{2}$$

where $C$ is the penalty factor regulating the tolerance for samples beyond insensitive zone, and $\ell_{\varepsilon}$ represents the $\varepsilon$-insensitive loss function in Equation (3):

$$\ell_{\varepsilon}(f(\mathbf{x}_i) - y_i) = \begin{cases} 0, & if \mid f(\mathbf{x}_i) - y_i \mid \le \varepsilon, \\ \mid f(\mathbf{x}_i) - y_i \mid -\varepsilon, & otherwise. \end{cases} \tag{3}$$

We introduce slack variables $\xi_i$ and $\xi_i^*$ to consider the training samples outside $\epsilon$-insensitive zone, then the Equation (3) can be rewritten as

$$\min_{\omega, b, \xi_i, \xi_i^*} \frac{1}{2}\|\omega\|^2 + C\sum_{i=1}^{m}(\xi_i + \xi_i^*), \tag{4}$$

$$s.t \begin{cases} f(\mathbf{x}_i) - y_i \le \varepsilon + \xi_i, \\ y_i - f(\mathbf{x}_i) \le \varepsilon + \xi_i^*, \\ \xi_i, \xi_i^* \ge 0, & i = 1, 2 \dots m. \end{cases} \tag{5}$$

By introducing a set of Lagrange multipliers, including $\mu_i \ge 0$, $\mu_i^* \ge 0$, $\alpha_i \ge 0$, $\alpha_i^* \ge 0$, a Lagrange can be created as

$$\begin{aligned} L(\omega, B, \alpha, \alpha^*, \xi, \xi^*, \mu, \mu^*) =& \frac{1}{2}\|\omega\|^2 + C\sum_{i=1}^{m}(\xi_i + \xi_i^*) - \sum_{i=1}^{m}(\mu_i\xi_i + \mu_i^*\xi_i^*) + \\ & \sum_{i=1}^{m}\alpha_i(f(\mathbf{x}_i) - y_i - \varepsilon - \xi_i) + \sum_{i=1}^{m}\alpha_i^*(y_i - f(\mathbf{x}_i) - \varepsilon - \xi_i^*). \end{aligned} \tag{6}$$

Let the partial derivativesfor $L$ with respect to $\omega$, $b$, $\xi_i$, and $\xi_i^*$ equal zero. The formula can be transformed into a dual problem and the solution is shown as follows:

$$f(\chi) = \sum_{i=1}^{m}(\alpha_i^* - \alpha_i)\mathbf{x}_i^T\chi + b. \tag{7}$$

After mapping into higherdimension space, Equation (7) can be expressed as

$$\begin{aligned} f(\chi) &= \sum_{i=1}^{m}(\alpha_i^* - \alpha_i)\kappa(\mathbf{x}_i, \mathbf{x}_j) + b \\ &= \sum_{i=1}^{m}\omega_i\kappa(\mathbf{x}_i, \mathbf{x}_j) + b, \end{aligned} \tag{8}$$

where $\kappa(\mathbf{x}_i, \mathbf{x}_j) = \phi(\mathbf{x}_i)\phi(\mathbf{x}_j)$ is the kernel function that converts the nonlinear cases into linear ones.

As one of the commonly used kernel functions, radial basis function (RBF) is used in this paper:

$$\kappa(\mathbf{x}_i, \mathbf{x}_j) = exp(-\frac{1}{2}(\frac{\|\mathbf{x}_i - \mathbf{x}_j\|}{\sigma})^2), \tag{9}$$

where $\sigma$ is the parameter of RBF, which controls the smoothness of kernel function varied by a positive infinite domain. The schematic view of SVR is shown in Figure 2. In this figure,

the input layer represents the environmental data, including pavement initial temperature, pavement average temperature, environment average temperature, humidity, radiation, wind speed, and water film thickness, while the output layer refers to the pavement icing time. It should be pointed out that the input variables limit the prediction accuracy; thus, it is necessary to optimize the input. RSM, a conventional method used to optimize the design parameters by manipulating the relationship between input and output variables, has been gradually used to improve the predict accuracy of SVR in recent studies [32]. Therefore, RSM is utilized in this paper to calibrate the input variables for SVR.

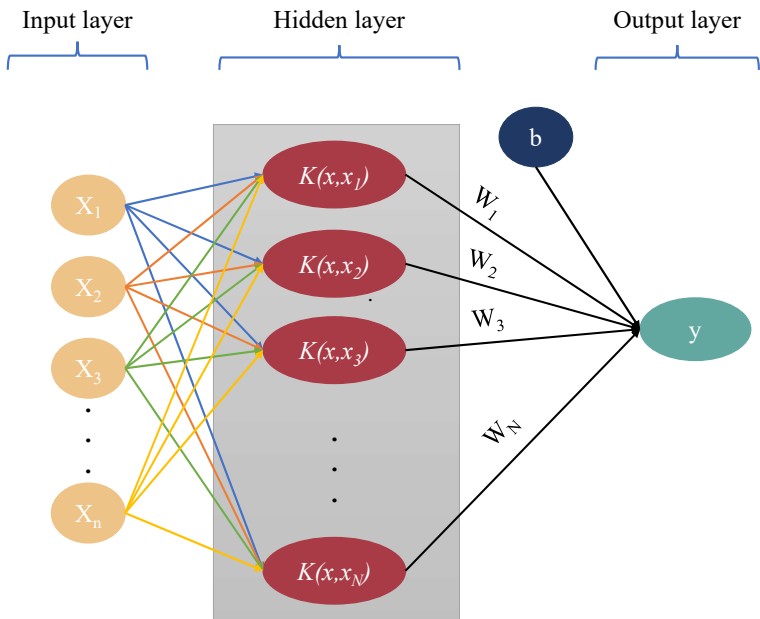

**Figure 2.** SVR structure.

*2.2. Response Surface Methodology*

Response surface methodology contains a collection of mathematical and statistical techniques used to develop the relationship between response and its related input factors. In general, the relationship is unknown, but can be approximated by a polynomial basis function of the form

$$r = f(\mathbf{q})\eta + \epsilon, \tag{10}$$

where $\mathbf{q} = (q_1, q_2, \ldots, q_k)^T$, $r$ is the response corresponding to $\mathbf{q}$, $f(\mathbf{q})$ is a vector consisting of the powers and cross products of powers of $q_1, q_2, \ldots, q_k$, $\eta$ is the coefficient corresponding to the element in $f(\mathbf{q})$, and $\epsilon$ is a random error that usually assumed to be zero. In this paper, a second-degree polynomial function is used, as shown in Figure 3, to calibrate the input variables for SVR. The formula of second-degree polynomial function is shown in (11):

$$r = \eta_0 + \sum_{i=1}^{n} \eta_i q_i + \sum_{i=1}^{n} \sum_{j=i}^{n} \eta_{ij} q_i q_j. \tag{11}$$

The weight coefficients of different components in Equation (11) are calibrated by the following equation [33]:

$$\eta = [P^T P]^{-1}[P^T O], \tag{12}$$

where $O$ is the output variables in the training datasets, $P = [1, q_1, q_2, q_1 q_2, q_1^2, q_2^2]$ for quadratic polynomial. The schematic view of RSM for calibrating hybrid improved SVR is shown in Figure 4. Total $k$ sets of environmental data in input layer1 are initially calibrated by RSM to produce input variables in input layer 2 using $m = \frac{n!}{2! \times (n-2)!}$, where $n$ is the number of environmental factors affecting the pavement icing time in each set

of environmental data, and *m* is the number of parameters after the calibration of RSM. Then, the input variables are used to calibrate the SVR model by kernel function. Since the parameters of the SVR model greatly affect the performance of prediction, it is necessary to obtain the optimal parameter combination for the pavement icing time prediction. The conventional approach for finding the optimal parameter combination is manual operation. This method takes a large amount of time and the final outcome may not be favorable. Considering its drawbacks, the PSO algorithm is used in this paper to calibrate the optimal SVR model.

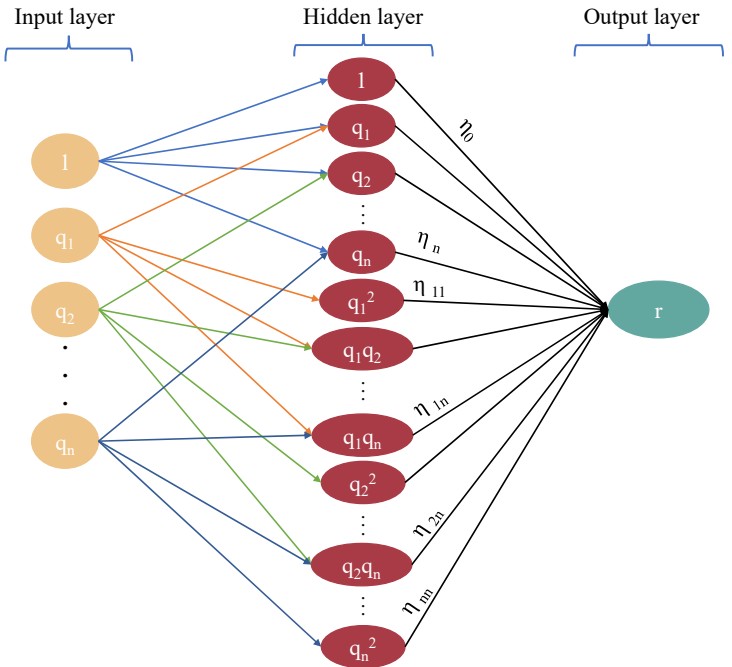

**Figure 3.** RSM schematic structure.

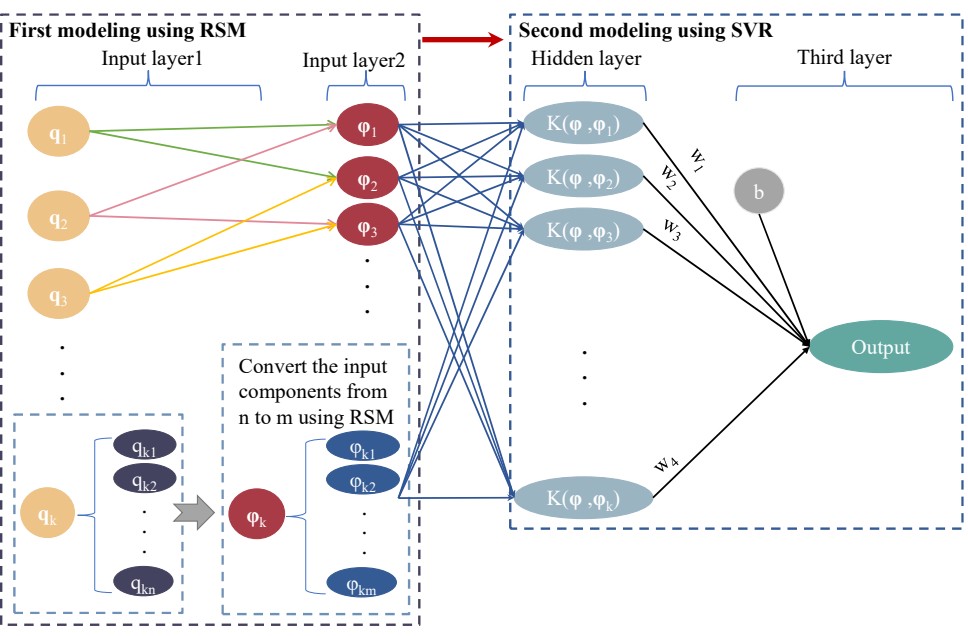

**Figure 4.** RSM-SVR framework.

*2.3. Improved Particle Swarm Optimization (PSO)*

PSO algorithm was first developed by Kennedy and James in 1995 based on a distributed behavior model [26]. This model is initialized with a swarm of particles in a D-dimension space, where D represents the number of parameters to be optimized. Each particle is composed of a speed vector ($V_i$) and a individual position vector in D dimensions. During the iteration, individual optimal position (*pbest*) of each particle is selected from individual position in the history and the global best optimal position (*gbest*) is determined among *pbest* for the whole swarm. To identify the optimal solution, each particle adjusts its flying according to the two optimal positions:

$$V_i^{t+1} = \omega^t V_i^t + c_1 r_1 (pbest_i^t - X_i^t) + c_2 r_2 (gbest^t - X_i^t), \tag{13}$$

$$X_i^{t+1} = X_i^t + V_i^{t+1}, \tag{14}$$

where $V_i^t$ denotes the speed of particle $i$ at iteration $t$, $X_i^t$ denotes the position of particle $i$ at iteration $t$, $pbest_i^t$ is the individual optimal position of particle $i$ at iteration $t$, $gbest^t$ is the global optimal position at iteration $t$, $c_1$ and $c_2$ are acceleration constant with the default value, 2, $r_1$ and $r_2$ are the random numbers ranging from 0 to 1, and $\omega^t$ is the inertia factor. Equation (13) contains three parts. The first part is called the momentum part, reflecting the flying inertia of the particle. The second part represents the particles approximating individual optimal position in the history, and the last part reflects the tendency to the global optimal position.

Previous studies have proved the performance of PSO in optimizing SVR parameters [34–36], while the local optimal solution has prevented the PSO to further optimize parameters after a certain number of iterations. Self-adaptive evolution strategy (ES) is a method used to enhance the operation of PSO when the global best position cannot be improved after some successive generations. This method applies small Gaussian mutations; thus, it is suitable for local optimization [37]. The Gaussian mutation is used by mutating the particle speed $V_i$ and inertia factor $\omega_i$:

$$V_i' = V_i exp[\tau' N(0,1) + \tau N_i(0,1)], \tag{15}$$

$$w_i' = w_i + V_i' N_i(0, \alpha T(g)), \tag{16}$$

$$T(g) = 1 - exp(-fitness(g)), \tag{17}$$

where $N_i(0,1)$ is the normal distribution for each particle, the value of $\tau$ and $\tau'$ are $(\sqrt{2n})^{-1}$ and $(\sqrt{2\sqrt{n}})^{-1}$, respectively, $N_i(0, \alpha T(g))$ represents the random value created by normal distribution, $\alpha$ is a positive proportional constant, $fitness(g)$ is the fitness value of individual, and $T(g)$ is the degree of mutation. After the resetting of velocity and inertia factor, *pbest* and *gbest* are evolved by self-adaptive evolution strategy. This has proved the diversity of the particles and enabled the particles to find a better optimal solution in the next iteration.

*2.4. Performance Indicators*

To evaluate the effectiveness of RSM and PSO methods in enhancing the prediction capability of SVR, four models, including SVR, PSO-SVR, RSM-SVR, and hybrid improved SVR, are trained and tested using the data obtained onsite, within which the hybrid improved SVR combines RSM and PSO to construct a prediction model. CV is used to divide the data into train set and test set and evaluate the prediction capability using four statistical comparative metrics, including the coefficient of correlation ($R^2$), mean absolute error ($MAE$), mean absolute percentage error ($MAPE$), and root mean square error ($MSE$). Generally, the $R^2$ indicates the correlation between variables, while the $MAE$, $MAPE$, and $MSE$ measure the difference between predicted values and actual values in an equal manner. Lower $MAE$, $MAPE$, and $MSE$ and higher $R^2$ correspond to higher prediction

accuracy and better agreement with actual data. These indicators are computed in terms of equations:

$$R^2 = \frac{\sum_{i=1}^{N}(\hat{y}_i - \bar{y}_i)}{\sum_{i=1}^{N}(y_i - \bar{y}_i)^2}, \tag{18}$$

$$MAE = \sum_{i=1}^{N}\frac{|\hat{y}_i - y_i|}{N}, \tag{19}$$

$$MAPE = \frac{1}{N}\sum_{i=1}^{N}\frac{|\hat{y}_i - y_i|}{y_i}, \tag{20}$$

$$MSE = \frac{\sum_{i=1}^{N}(\hat{y}_i - y_i)^2}{N}, \tag{21}$$

where $\bar{y}_i = \frac{1}{N}\sum_{i=1}^{N}y_i$, $N$ is the amount of test data. Besides that, the PSO-SVR and proposed algorithm are also compared from the change of fitness to evaluate the convergence speed.

## 3. The Application Results and Discussion

### 3.1. The Raw Data of Pavement Icing Information

The hybrid improved SVR is applied in the pavement icing prediction and compared with the SVR, PSO-SVR, and RSM-SVR to evaluate its performance. A total of 188 sets of data, including icing time, pavement initial temperature, pavement average temperature, environment average temperature, humidity, radiation, wind speed, and water film thickness, are obtained onsite for the prediction. CV method is used in this paper to separate the input data into training set and test set. To guarantee the reliability of the prediction model, 80% of the data are selected as training set and the remaining 20% are used to verify the accuracy of the model. The details can be seen in Table 2.

**Table 2.** Pavement icing information.

| Item | Data Set | | | | |
|---|---|---|---|---|---|
| | 1 | 2 | 3 | ... | 188 |
| Icing time (min) | 43 | 40 | 45 | ... | 16 |
| Pavement initial temperature | −2 | −1.31 | −1.68 | ... | −4.5 |
| Pavement average temperature | −0.75 | −0.71 | −1.09 | ... | −3.52 |
| Environment average temperature | −3.27 | −2.49 | −2.24 | ... | −8.4 |
| Humidity | 58.25% | 60.24% | 60.78% | ... | 62.06% |
| Radiation | 0 | 0 | 0 | ... | 0 |
| Wind speed | 0 | 0 | 0 | ... | 0.39 |
| Water film thickness (cm) | 1 | 0.8 | 2.4 | ... | 1.8 |

### 3.2. Comparison and Analysis of Accuracy

Figure 5 illustrates the different scatter plots of the train set and test set for the prediction value of four models and the true value of the measured icing time. In the figure, the $y = x$ line is plotted to represent the best performance, and a confidence zone is formed by 15% and −15% error line. It shows that fewer data points fall beyond the confidence zone ranging from −15% to 15% for both train set and test set after SVR is improved by RSM or PSO. RSM-SVR shows better performance compared with PSO-SVR, for which 52 data points are in the train set and 25 data points are included in the confidence zone. In RSM-SVR, the value of $R^2$ for train set and test set is, respectively, 0.9258 and 0.8771, which sees an enhancement of 2.73% and 1.74% compared with PSO-SVR. It is concluded that the calibration of input variables by RSM will outperform the effect of PSO if the parameter combination, $\sigma$ and $C$, is already approximated to the optimal solution.

Moreover, the combination of two methodologies appears to demonstrate best optimization effects with 61 in train set and 31 in test set, and the $R^2$ are, respectively, 0.9655 and 0.9318. It indicates the availability of coupling these two methods for optimization.

In Figure 6, the four models are compared for *MAE*, *MAPE*, and *RMSE*. The original *MAE*, *MAPE*, and *RMSE* for SVR are, respectively, 10.7087, 2.6735, and 0.2116 in training and 9.8709, 2.5748, and 0.2038. After the application of RSM or PSO, the three items all see an enhancement, which indicates the effectiveness of the two methodologies in improving SVR. This trend is similar to the results in past research [27,31]. It suggests that RSM and PSO can both contribute to the improvement of prediction accuracy of SVR, while RSM demonstrates better performance than PSO, for which the three indexes are, respectively, 7.4895, 2.2990, and 0.1833 in training and 8.2087, 2.2472, and 0.1642 in testing. This demonstrates that the optimization of input variables can achieve better performance in enhancing the prediction capability of SVR compared with optimizing the SVR parameters. In addition, the combination of two methodologies obtains the best optimization result with 3.4849, 1.3437, and 0.1032 in training and 4.5519, 1.7646, and 0.1233 in testing. More details are shown in Table 3.

Figure 7 displays the predictive value of four models and the true value in the test set. According to the ratio set in CV method, 38 sets of data are used in this section to verify the capability of four models established in this paper. The comparison demonstrates that predictive value of hybrid improved SVR, PSO-SVR, and RSM-SVR is more approximated to the true value than the original SVR. The data points of PSO-SVR are more deviated from the true value in contrast to RSM-SVR, while the result is further improved as the two optimization methodologies, RSM and PSO, are combined to optimize SVR. This corresponds to the results in Figure 5 and Table 3.

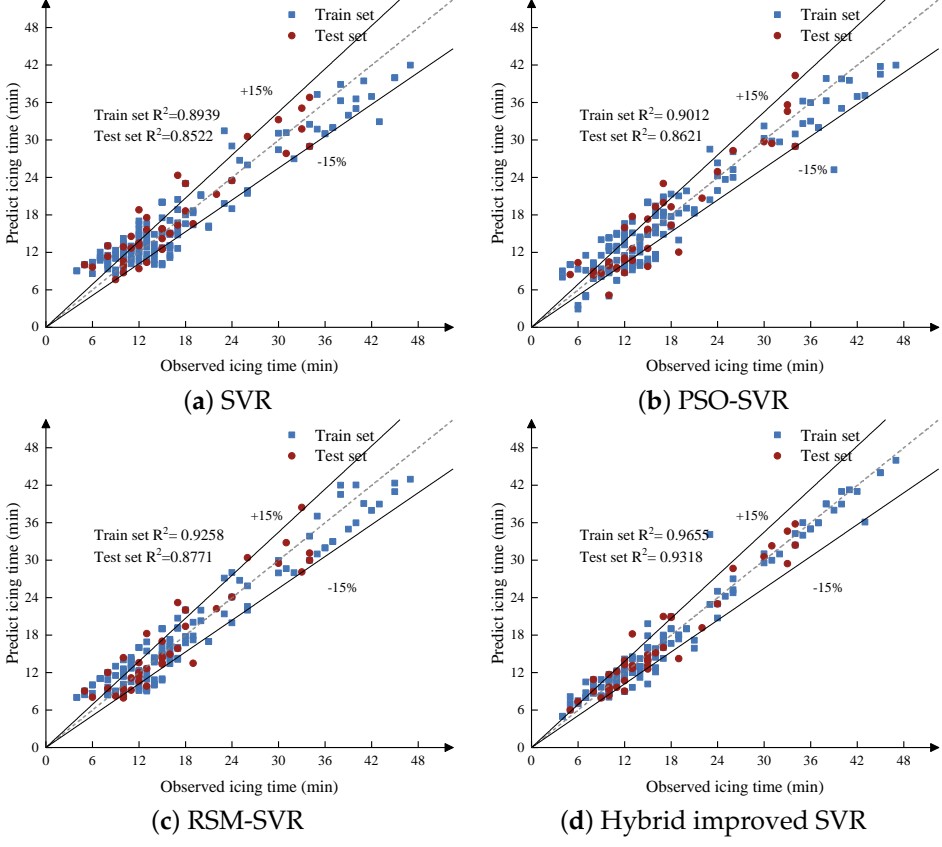

**Figure 5.** The correlation coefficient.

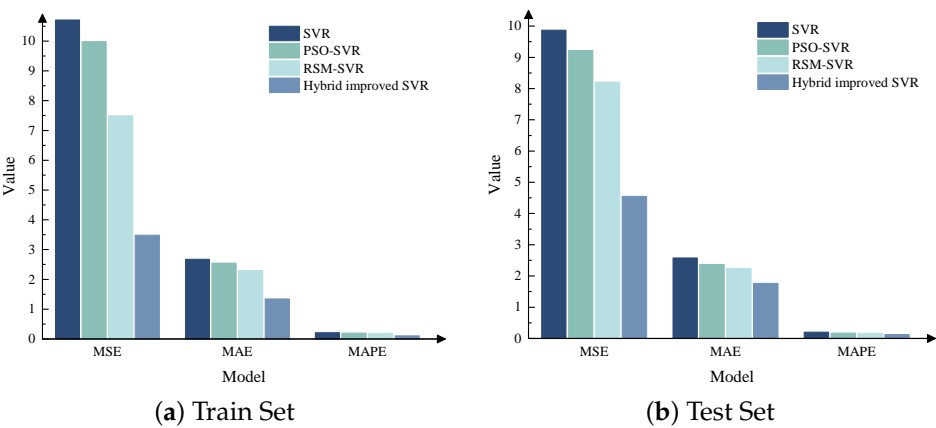

**Figure 6.** The comparison of accuracy.

**Table 3.** The accuracy data of three models.

| Item | Train Set | | | | Test Set | | | |
|------|-----------|---------|---------|-----------------------|----------|---------|---------|-----------------------|
| | SVR | PSO-SVR | RSM-SVR | Hybrid Improved SVR | SVR | PSO-SVR | RSM-SVR | Hybrid Improved SVR |
| MSE | 10.7087 | 9.9797 | 7.4895 | 3.4849 | 9.8709 | 9.2145 | 8.2087 | 4.5519 |
| MAE | 2.6735 | 2.5443 | 2.2990 | 1.3437 | 2.5748 | 2.3709 | 2.2472 | 1.7646 |
| MAPE | 0.2116 | 0.2003 | 0.1833 | 0.1032 | 0.2038 | 0.1713 | 0.1642 | 0.1233 |

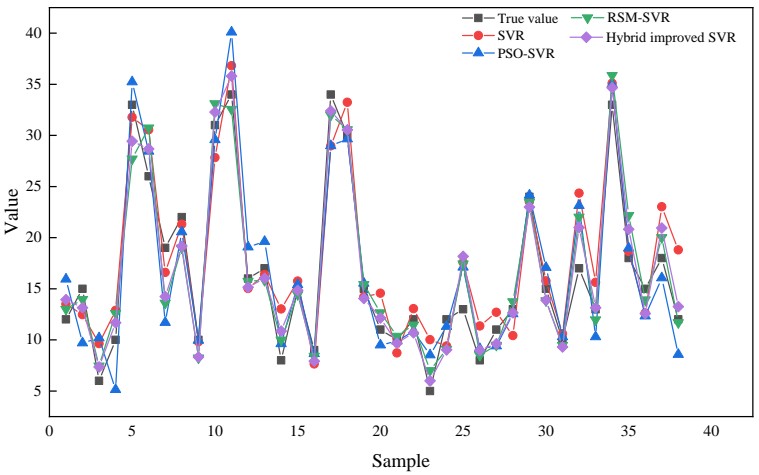

**Figure 7.** The prediction capability of different models.

*3.3. Comparison and Analysis of Convergence Speed*

The hybrid improved SVR and PSO-SVR utilize PSO to optimize the SVR kernel parameters; thus, the convergence speeds of the two models are compared with each other. In this paper, $R^2$ is defined as the fitness of PSO to change the combination of $\sigma$ and $C$ for the optimal solution in the iteration. The fitness change from the initial age to the end is displayed in Figure 8. It can be observed that the fitness value at the beginning stage is approximately 0.87 and 0.857 for hybrid improved SVR and PSO-SVR. The higher fitness value of hybrid improved SVR at the first iteration age is due to the calibration process of RSM that increases the number of input variables by polynomial expression. As the iteration continues, the two methods both see a rise of fitness value with the functionality of PSO. Around the age of 20, the two methods attain their optimal fitness, and successive minor change is observed before that because of the slight adjustment of self-adaptive evolution strategy. This means that the similar convergence speed is obtained after PSO is combined with RSM to calibrate SVR, compared with PSO-SVR.

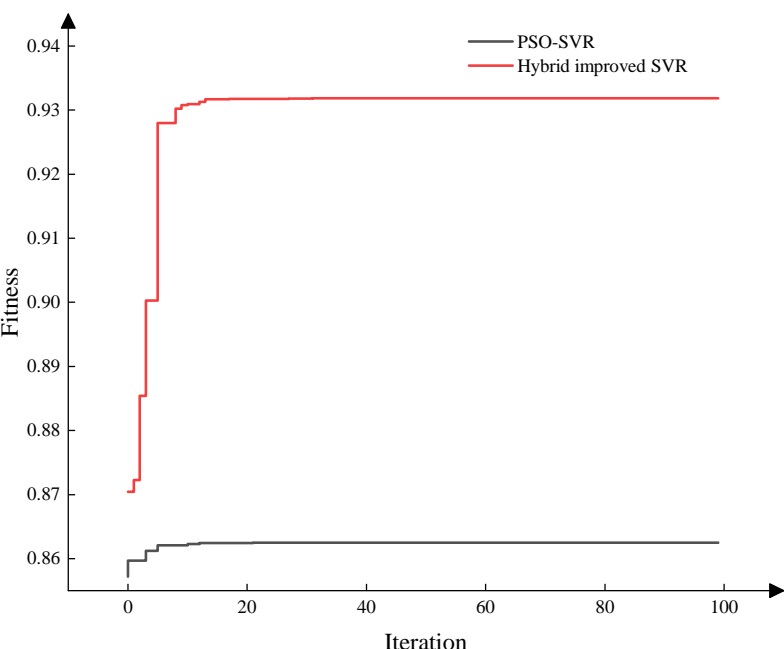

**Figure 8.** Fitness change.

## 4. Conclusions

Pavement icing is a critical factor affecting traffic safety and efficiency, and it is difficult to predict icing time due to the complexity of freezing. Support vector regression is a newly developed algorithm with the advantages of small-scale sample, high accuracy, and easier training. Thus, it is suitable for predicting icing time. This paper investigates the utility of RSM and self-adaptive ES-based PSO for calibrating SVR in order to obtain a hybrid improved model, applied for predicting icing time. A total of 188 sets of environmental data and the corresponding icing time collected onsite were divided into train set and test set by CV method using a train set percentage of 80%. The input variables of SVR were generated by RSM using the environmental data in train set, and the input variables were combined with corresponding icing time data to construct the original SVR model. The adopted improved PSO was used to optimize the penalty coefficient $C$ and kernel function parameter $\gamma$ to obtain the hybrid improved SVR model. To evaluate the performance of the proposed model, the test set was used to verify the accuracy of model through different indicators, including $R^2$, $MAE$, $MAPE$, and $MSE$. Besides that, the convergence speeds were used to compare the convergence capability between the hybrid improved SVR and PSO-SVR. It indicates that the convergence speed remains constant while the accuracy of SVR is greatly improved with the combined optimization of RSM and PSO.

The proposed algorithm provides an alternative in predicting pavement icing time and it is helpful to the road transportation system. The algorithm considers the input variables and the model parameters and greatly enhances the prediction performance of the model with the combination of RSM and PSO, which signifies a new approach to improve SVR. Meanwhile, the calculation consumption and the complexity of the proposed model is relatively high compared with the SVR, without the application of optimization methodology, due to the iteration of PSO and the calibration of RSM, which hinders its application in long-term pavement icing time prediction. We are conducting research to reduce the calculation consumption of SVR while keeping the accuracy relatively high by simplifying the composition of PSO. Besides that, related work is also conducted to investigate the approach for the calibration of SVR input variables with simple composition.

**Author Contributions:** Conceptualization, Z.D.; data curation, W.X.; formal analysis, L.S.; funding acquisition, Z.D., Q.Z. and Q.L.; investigation, W.X.; methodology, L.S., W.X. and Y.Y.; project administration, Q.Z. and Q.L.; resources, W.X., Q.Z. and Q.L.; software, L.S. and Z.D.; supervision, Y.Y.; validation, L.S., Q.Z. and Q.L.; visualization, Y.Y. and W.X.; writing—original draft, L.S.; writing—review and editing, Y.Y., Q.Z. and Q.L. All authors have read and agreed to the published version of the manuscript.

**Funding:** This work is supported by the project "Early Warning and Rapid Processing Technology for Expressway Pavement Icing".

**Institutional Review Board Statement:** Not applicable.

**Informed Consent Statement:** Not applicable.

**Data Availability Statement:** Not applicable.

**Conflicts of Interest:** The authors declare no conflict of interest.

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
