# Peer review of "Icing Time Prediction Model of Pavement Based on an Improved SVR Model with Response Surface Approach"

_applsci, doi:10.3390/app12168109_

Round 1

Reviewer 1 Report

1. The authors may check the word limits for abstract. It seems to be quite lengthy.

2. Is SVM a new approach as the authors claim in the abstract? SVM has been widely used in research for a long time. The authors may clearly explain the exact model they have proposed in the study for better understanding of the readers.

3. The methodology and results sections are well-written. However, the authors may need to discuss their findings in the light of past research and highlight what are the similarities and contradictions between the current and past findings. From there, the authors can summarize the contributions and novelty of the current work.

4. The literature review in section can be better represented in the form of a table where the authors can summarize the past research and highlight their limitations in one of the columns. This will help the authors in providing clarity of the research gaps which current study aims to work on.

5. There are no limitations of the study mentioned in the last section. No study is complete without illustrating the limitations observed during the conduct of the study. The authors need to mention them in their conclusions.

6. The conclusions of the study can be better represented in bullets signifying the takeaways to the policymakers and future researchers.

Overall, the manuscript is well-designed and well-written, however the authors can further enhance it by incorporating the above suggestions. Best wishes to the authors.

Reviewer 2 Report

This article presents an improved hybrid SVR algorithm to predict pavement icing time with an improvement operation using the response surface method (RSM) and particle swarm optimisation (PSO). The experimental results show that RSM and PSO affect promoting accuracy. In contrast, combining the two methods in an improved hybrid algorithm has shown a better optimisation capacity ( PSO-SVR, RSM-SVR and SVR ).

The current article complies with what is necessary to be accepted, however I would add the following:

i) Justify why only this method is used and comparison with others.

ii) Include a session of related work.

iii) Indicate the source of data and codes used, in some repository accessible to replicate the experiments.

Reviewer 3 Report

This study presented a modified model - a combination of SVR and RSM for prediction of pavement icing time.  The proposed model performances was compared with the SVR, PSO-SVR and RSM-PSO using some performance measures. There are a few comments/ clarifications needed from the author as listed below:

1. In the Introduction, the author paid too many attention on describing the pro and cons of the model, however very lack literatures were presented on the previous models used to model pavement icing time. In addition, no exposure on the parameters that affects the pavement icing time was describe in the introduction.

2. The method was the main weakness in this manuscript. Very confusing flow of method and lack of details were found out in this section. For example, for the modified model, various term were used e.g. modelling method, hybrid model or modified model. It was not properly highlighted that this were the proposed model in this manuscript. The parameters used as inputs were not explained. In addition, the abbreviation for each model e.g. y, b, r etc were not explained.

3. The results were briefly explained and the discussion was very brief. It was stated in Section 3.1 that only 20% of the data were used for training and the rest was for test. Is this correct? No critical discussion was given related to the proposed model or the well-known model in the field of pavement icing time.

4. The conclusion merely summarized the methods and results which was not appropriate. The author should relate the findings with the theory or application on the specific field of research.

The details of the comments were given in the attached document.

Reviewer 4 Report

The manuscript applied a hybrid SVR (PSO-SVR) model for the prediction of pavement icing time. Total of 40 samples was used to train and test models (if I am wrong, the authors must describe the implemented data sufficiently.  The number of data to train machine learning models is too low. So, the findings are questionable. Moreover, the authors merely used 20% of the data to train the modes (line 190). This may result in incorrect models. At least 70% of data is suggested to train the models. Recent studies on SVR showed that can be perfectly Improved by FFA (see [1,2]) even better than PSO. Comparing FFA, explain why you suggest PSO.

1- Danandeh Mehr, A., Nourani, V., Karimi Khosrowshahi, V., & Ghorbani, M. A. (2019). A hybrid support vector regression–firefly model for monthly rainfall forecasting. International Journal of Environmental Science and Technology16(1), 335-346.

2- Yang, H., Nikafshan Rad, H., Hasanipanah, M., Bakhshandeh Amnieh, H., & Nekouie, A. (2020). Prediction of vibration velocity generated in mine blasting using support vector regression improved by optimization algorithms. Natural Resources Research29(2), 807-830.

Round 2

Reviewer 3 Report

Overall, all the comments in the previous review had been addressed well by the authors.

Reviewer 4 Report

The authors well addressed my comments.